# Cyclic di-GMP Modulation of Quorum Sensing and Its Impact on Type VI Secretion System Function in *Sinorhizobium fredii*

**DOI:** 10.3390/microorganisms13102232

**Published:** 2025-09-24

**Authors:** Juan Aranda-Pérez, María del Carmen Sánchez-Aguilar, Ana María Cutiño-Gobea, Francisco Pérez-Montaño, Carlos Medina

**Affiliations:** Departamento de Microbiología, Facultad de Biología, Universidad de Sevilla, Avda. Reina Mercedes 6, CP41012 Sevilla, Spain; juaaraper@alum.us.es (J.A.-P.); msanchez44@us.es (M.d.C.S.-A.); acutino@us.es (A.M.C.-G.); fperezm@us.es (F.P.-M.)

**Keywords:** quorum sensing (QS), *N*-acyl homoserine lactones (AHLs), cyclic di-GMP (c-di-GMP), Type VI Secretion System (T6SS), *Sinorhizobium fredii*, biofilms, symbiosis, nodulation, legumes

## Abstract

Effective rhizobium–legume symbiosis depends on multiple molecular signaling pathways, integrating not only classical nodulation factors and surface polysaccharides but also diverse protein secretion systems. Among them, the Type VI Secretion System (T6SS) has emerged as a key player, due to its dual roles in interbacterial competition and interactions with eukaryotic hosts, though its contribution to symbiosis remains unclear. Key regulatory messengers, including the main autoinducer of the quorum sensing (QS) systems, the *N*-acyl homoserine lactones (AHLs), and the second messenger cyclic di-GMP (c-di-GMP), modulate the transition between motility and biofilm formation, especially in the context of bacteria interacting with eukaryotes, including rhizobia. While c-di-GMP’s impact on exopolysaccharide production in these organisms is well established, its influence on protein secretion systems, particularly in conjunction with QS, is largely unexplored. To contribute to the study of such interplay, we artificially increased intracellular c-di-GMP levels by overexpressing a heterologous diguanylate cyclase in three *Sinorhizobium fredii* strains of agronomic relevance. This engineering revealed strain-specific outcomes, since elevated c-di-GMP enhanced biofilm development in two strains, but reduced it in another. Furthermore, using β-galactosidase expression assays, we confirmed that both high c-di-GMP and/or AHL concentrations contribute to the transcriptional activation of T6SS. These results demonstrate a direct regulatory link between c-di-GMP, QS signals, and T6SS expression, shedding light on the multilayered control mechanisms that structure beneficial rhizobia–plant interactions.

## 1. Introduction

The study of bacterial communication and signaling mechanisms has garnered significant attention due to its implications in various biological processes and potential applications in biotechnology and medicine. Among the diverse bacterial species where these processes have been studied, rhizobial strains stand out for their symbiotic relationship with leguminous plants, where they play a crucial role in nitrogen fixation [1]. This symbiosis results in the formation of root nodules, where rhizobia fix atmospheric nitrogen into a form that plants can utilize. This process is vital for plant growth and soil fertility, making rhizobia an essential component of sustainable agriculture [2]. Beyond its practical applications, the rhizobium-legume symbiosis provides an important model for investigating complex and intimate interactions between bacteria and eukaryotic hosts. The specificity of this association is determined by the exchange of molecular signals between both partners, enabling accurate recognition and the successful initiation of the symbiotic relationship [1]. Rhizobial signals traditionally studied were the nodulation factors, synthesized when the flavonoids secreted by legumes interact with the rhizobial NodD protein that triggers the expression of nodulation (*nod*) genes whose products are involved in the synthesis of lipo-chito-oligosaccharides (LCOs). In turn, bacterial LCOs are perceived by plants receptors triggering the nodulation process through a complex plant signaling process [1,2]. However, these molecules are not the only signals governing symbiotic specificity. Surface polysaccharides such as cyclic glucans, lipopolysaccharides, K-antigen capsular polysaccharides, and/or exopolysaccharides are also essential contributors. These molecules act both as signals that facilitate the development of the symbiosis and as protective agents that help bacteria evade plant defense mechanisms [3]. Additionally, some rhizobial strains, but not all, use protein secretion systems such as the Type III (T3SS), Type IV (T4SS), and Type VI (T6SS), to deliver effector proteins into host cells. One of the main functions of these effectors is to suppress plant defense responses to facilitate bacterial infection and survival inside the host, completing the plethora of molecules involved in the signaling of the process reviewed in Jiménez-Guerrero et al. [4]. Specifically, the T6SS delivers effectors or toxins directly into both prokaryotic and eukaryotic cells. Therefore, this mechanism is remarkably important for interbacterial competition when directed against other bacteria in shared environments, while in eukaryotic hosts, T6SS contributes to successful infection [5]. The precise functions of T6SS effectors in rhizobial symbiosis remain poorly understood, although some studies indicate that these proteins may have neutral, beneficial, or detrimental impacts depending on the specific symbiotic partnership [6,7,8,9,10,11].

Due to the diversity of all these signal molecules and proteins involved in the establishment of a successful symbiotic interaction, their synthesis must be coordinated from an upper tier that probably implicates master regulatory genes whose function are controlled by common molecules interconnected in a complex network with the metabolic fitness of the bacteria. In rhizobia and other soil bacteria, lifestyle shifts from motile to biofilm-forming units are governed by environmental cues and changes in the intracellular concentration of nucleotide-based second messengers like bis-(3′-5′)-cyclic guanosine monophosphate (c-di-GMP) [12]. Hence, a high c-di-GMP concentration generated by the action of diguanylate cyclases (DGC) regulate processes associated with sessility promoted by the production and secretion of adhesins and exopolysaccharides that conforms the biofilm matrix, while the motile and in some cases virulent way of life of soil bacteria is generally related with low levels of c-di-GMP produced by the action of phosphodiesterases (PDE) [12]. Although the implication of c-di-GMP in cell signaling and gene regulation in the virulence of plant pathogens has been well documented [13], its implication in the symbiotic performance of rhizobia is still not well understood, with scarce data available. Thus, while the influence of c-di-GMP on extracellular polysaccharide biosynthesis in rhizobia has been thoroughly investigated and described [14,15], the precise regulation of protein secretion systems by c-di-GMP in rhizobia and other beneficial plant-interacting bacteria is not well understood and warrants further investigation [16]. In this sense, the synthetic increment of c-di-GMP concentrations triggered by the overexpression of PleD (a DGC from *Caulobacter crecentus*) has been recurrently used as a tool for study the generation of rhizobial exopolysaccharides [17,18].

On the other hand, soil bacteria use to coordinate their behaviour and gene expression based on population density through a mechanism named Quorum sensing (QS) by which individual bacteria “sense” how many of their peers are around by releasing and detecting small and diffusible chemical signals called autoinducers of different nature [19]. The most extended autoinducers among Gram-negative bacteria are *N*-acyl homoserine lactones (AHLs), whose composition varies in their fatty acid chain length and functional groups of the third carbon, which influences their specificity. Therefore, QS is a cell-to-cell communication mechanism that enables bacteria to collectively modify and synchronize behaviors, including those crucial for interactions with eukaryotic hosts. In rhizobia, QS-controlled phenotypes seem to play a role during free-living conditions whereas others are involved in symbiotic performance with legume hosts. In the model bacterium *Sinorhizobium meliloti*, these systems regulate a range of functions, such as plasmid transfer, polysaccharide production, motility, growth rate, and nodulation [20].

Combining and contextualizing these networks, QS and c-di-GMP-dependent gene regulation are interconnected in many bacterial species, including with mechanisms that involve the regulation of c-di-GMP biosynthesis or degradation by QS, leading to reduced c-di-GMP levels in the QS state [21]. Conversely, in *S. meliloti*, QS appears to have no impact on the overall cellular concentration of c-di-GMP. In fact, both normal and increased levels of c-di-GMP suppress the expression of the AHL synthase gene *sinI* and reduce AHL accumulation in the growth medium [22].

However, to date the involvement of c-di-GMP and AHLs in the regulation of T6SS in rhizobia remains unexplored. To approach some of these intriguing questions, in this work we use several *Sinorhizobium fredii* strains that have a remarkable interest for their interesting properties. *S. fredii* USDA 257 (hereafter USDA257) is one of the rhizobia with a broader host range along with *S. fredii* NGR234 and *S. fredii* HH103, but with the particularity that presents two T3SS and one T6SS, while the others only have T3SS’s [11]. The implication of the T6SS in the symbiotic process in USDA257 is still under analysis by our group, and its particularities are distant from other rhizobial strains. *S. fredii* SMH12 (henceforth SMH12), is an interesting strain that has been recently sequenced and to whose study of QS mechanisms our group has previously contributed. In this strain, the development of a microcolony-type biofilm controlled by QS is crucial for successful root colonization and optimal symbiosis with soybean [23,24,25]. And lastly, *S. fredii* HWG35 (from here HWG35) is a highly competitive strain that was isolated from nodules of soybean cultivars in Wei Fang County, Shandong Province, China [26]. Although it seems closely related to USDA257 and SMH12, HWG35 is also capable of nodulate American soybeans [27], a feat that these two other strains cannot achieve, making it an interesting subject for the future study of protein secretion involved in nodulation. The striking behavioral differences observed among these phylogenetically close strains under identical experimental conditions point to diverse regulatory mechanisms, thereby establishing this system as an attractive model for further molecular and functional studies.

## 2. Materials and Methods

### 2.1. Bacterial Strains and Plasmids

Bacterial strains and plasmids used in this study are listed in Table 1. Rhizobial strains were cultured at 28 °C using either tryptone yeast (TY) medium [28] or yeast extract mannitol (YM) medium [29], adding 2% agar for solid media. When necessary, antibiotics were added to the media as previously described [30]. Genistein was prepared in ethanol and applied at a concentration of 1 μg/mL, resulting in a final concentration of 3.7 μM, and was used when needed. *Escherichia coli* strains were cultivated on LB medium [31] at 37 °C using 0.3 mM diaminopimelic acid (DAPA) as supplement. *Agrobacterium tumefaciens* NT1 (pZLR4) was grown at 28 °C on YM. Commercial AHLs were dissolved in methanol and used at various concentrations. Both genistein and AHLs were procured from Fluka (Sigma-Aldrich, St. Louis, MO, USA).

Triparental matings, as described in [17], were employed to deliver the mini-Tn7 constructs into the genomes of the three *S. fredii* strains. This transposon precisely integrates into attTn7 site of the conserved *glmS*. In USDA 257 and SMH12, only one copy of *glmS* seems to have an attTn7 site, according to known attTn7 sequences from different bacteria. *E. coli* β2163 harboring the pUX-BF13 plasmid carrying the transposase genes was used as a helper strain for transposition. Mini-Tn7 plasmids carrying the *pleD** gene were propagated in *E. coli* β2155 (*lacI^q^*) to avoid overexpression of this gene. Control plasmids with empty transposons were maintained using the *E. coli* β2163 strain. The mini-Tn7 insertions in each bacterial strain were verified by PCR using combinations of primers pTn7-R (CACAGCATAACTGGACTGATTTC), pTn7-L (ATTAGCTTACGACGCTACACCC), glmS-R (GGATTTCCCGCCAAGACGC) and glmS-L (CCCGTCATCGTCATTGCGCC). To verify the *pleD* insertion phenotype and to observe the production of exopolysaccharides, strains were grown on solid TY supplemented with Congo red (CR; 125 μg/mL), that binds to extracellular polysaccharides turning bacterial biomass color to red.

### 2.2. Thin Layer Chromatography Analysis (TLC)

Supernatants of the *S. fredii* strains grown during six days in 5 mL of YM medium (TY for SMH12). For USDA257 and SMH12, 500 µL of culture (1 mL for HWG35) were extracted with dichloromethane (50:50 *v*/*v*), evaporated to dryness, resuspended in 15 µL of ethyl acetate and analyzed by TLC, (HPTLC plates RP-18 F254s 1.13724, Merck, Darmstadt, Germany) using methanol:water (60:40 *v*/*v*) as eluent, dried and developed with *A. tumefaciens* NT1 (pZLR4) for the detection of AHLs as described previously [23]. A 4 µL (2 + 2) mixture of commercial AHLs (50 mg/mL C6-HSL and 5 mg/mL C8-HSL) was used as control [24].

### 2.3. β-Galactosidase Activity Determination

Two different β-galactosidase assays were conducted to: (i) evaluate AHL production in relation to varying c-di-GMP levels, and (ii) investigate T6SS regulation under different conditions. For quantifying total AHL production at different c-di-GMP concentrations, *A. tumefaciens* NT1 (pZLR4) served as a biosensor, following the protocol described by Pérez-Montaño et al. [23] with minor adjustments. A 10% (*v*/*v*) concentration of supernatant from cultures of each strain grown for 6 days in either YM or TY medium was used. 7.5 μL of C6-HSL (0.5 mg/mL) was included as a positive control, while fresh media served as the negative control.

To examine T6SS gene expression in USDA257 under different conditions, the plasmid pMUS1483 carrying the *ppkA* promoter fused to *lacZ* was introduced into both USDA257 mini-Tn7 and USDA257 mini-Tn7*pleD** strains. Transconjugants were cultured in 3 mL of TY medium for either 24 or 72 h at 28 °C with shaking at 180 rpm. To assess the influence of QS, 10% (*v*/*v*) supernatant from 6-day TY-grown bacterial cultures was added to the medium. β-galactosidase activity was measured according to the method of Miller with minor modifications [37]. In short, 24- or 72-h cultures of USDA257 strains harboring the pMUS1483 plasmid containing in each case the supernatants described above were collected, permeabilized and incubated with ONPG until the reaction turned the media color to pale yellow. Absorbance at 420 nm was measured and relativized to OD_600_ of cultures to calculate Miller units.

All experiments were carried out at least three times, each with three biological replicates and two technical replicates per condition, the standard errors of the mean were calculated.

### 2.4. Biofilm Formation Assays and Flocs Visualization

The initial stage of bacterial biofilm formation involves the attachment of bacteria to either biotic or abiotic surfaces. In this study, bacterial adhesion to each other (floculation) was analyzed by direct observation since clumps impeded accurate measurements, whereas binding to an abiotic surface (polystyrene), which is indicative of biofilm formation, was evaluated using the microtiter plate assays method developed by O’Toole and Kolter [38] with some modifications. Briefly, bacterial strains were first cultured in 5 mL of the appropriate medium and grown for 72 h at 28 °C. Each culture was then diluted to an OD_600_ of approximately 0.2, and the diluted suspensions were added to U-bottom polystyrene microtiter plates (Deltalab S.L., Rubí-Barcelona Spain). The plates were incubated at 28 °C with gentle shaking (100 rpm) for 6 days. After incubation, non-adherent cells were carefully removed, and the plates were dried, leaving only the bacteria that remained attached as an indicator of biofilm formation. The plates were subsequently rinsed three times with 0.9% NaCl and dried again. Then, 100 μL of 0.1% crystal violet solution (in water) was added to each well, incubated for 20 min, and then the wells were washed three times with distilled water by soaking for 30 s each time. After another drying step, 100 μL of 96% ethanol was added to each well, and the absorbance at 570 nm was measured using a Synergy HT microplate reader (BioTek, Seattle, WA, USA) using standard parameters. Experiments were performed a minimum of three times, with each run including three biological replicates and two technical replicates for every condition. For each wild-type strain, the absorbance at 570 nm was normalized to one.

### 2.5. Statistical Analysis

All statistical analyses were performed using GraphPad Prism 8.0.2 (GraphPad, Boton, MA, USA), with details provided in the figure legends.

## 3. Results

### 3.1. High Levels of Cyclic di-GMP Reduce AHL Production in S. fredii

The connection between the c-di-GMP molecular signaling network and regulatory mechanisms derived from population density modifications via QS has been previously described in various bacterial species [21]. To analyze if this interweaving exists in the different *S. fredii* strains analyzed, and if it also relates to the presence of inducing flavonoids (which regulate the transcription of numerous genes via NodD), we introduced the mini-Tn7*ple*D* transposon (and its empty homolog) into the *S. fredii* strains USDA257, HWG35, and SMH12, obtaining conjugation frequencies between 10^−8^ and 10^−9^, as previously described in other *S. meliloti* strains [17]. SMH12 was tagged with the Tc version of the transposon, since after several attempts we could not introduce the Km version as in the other strains. The resulting tagged strains, hereafter referred to as PleD, showed differential growth patterns in different media that will be discussed below. Subsequently, we performed modified β-galactosidase activity assays based on AHL concentration to detect alterations in the overall production of AHL [23]. For this, filtered supernatants obtained after incubating both the wild-type and PleD versions of USDA257, HWG35, and SMH12 in the presence or absence of genistein for 6 days (late stationary phase) were added to a final concentration of 10% to a culture of *A. tumefaciens* NT1 (pZLR4). This strain is used as a biosensor to analyze the capacity of the supernatants to induce β-galactosidase activity with a wide range of AHL [34]. Thus, β-galactosidase activity levels are proportional to AHL concentration in the supernatants collected in the stationary phase and are also indicative of the QS cellular state. The presence of high levels of c-di-GMP decreases AHL production in all strains, as observed by the reduction in β-galactosidase activity (Figure 1). Specifically, production was reduced by 20% in HWG35-PleD and by 50% in USDA257-PleD. Most remarkably, SMH12-PleD showed a reduction around 95% in AHL production (Figure 1A). This finding is striking, as this strain consistently demonstrated the highest AHL levels in this and previous assays [23,24]. The addition of genistein also showed a different behavior among these strains. While no appreciable increase in AHL production was observed in USDA257 in the presence of genistein, the HWG35 strain increased by about 30% (Figure 1A), similar to the behavior of SMH12 in previous observations [23]. A subtle but not significant restoration in AHL production in the presence of genistein was observed for the USDA257-PleD strain, achieving 75% of the wild-type level. Conversely, genistein supplementation failed to recover AHL production in the HWG35-PleD and SMH12-PleD strains. This highlights a powerful role for c-di-GMP in suppressing AHL production for intercellular communication, a suppression not overcome by flavonoid addition in these particular strains.

The β-galactosidase assay quantifies AHL production but does not reveal the specific chemical nature of these molecules. Thus, to verify if the increase in c-di-GMP concentration modifies the spectrum and variety of these molecules, we performed TLC assays that allow us to observe the chromatographic mobility of the produced molecules using as biosensor strain *A. tumefaciens* NT1 (pZLR4), which detect a wide-range of AHLs (fatty acyl chains ranging from C4 to C14 with the three functional groups). A 1:1 mixture of commercially acquired C6 and C8-AHLs was used as a control. As is displayed in Figure 1B, while an increase in c-di-GMP did not alter the general AHL composition for each strain, it affected their concentration, being massive the reduction in case of SMH12. Based on the relative mobility’s of control AHLs and prior findings for the SMH12 strain, TLC analysis indicates that all three bacteria produce at least 3-oxo-C8-HSL (which shows intermediate mobility between C6- and C8-HSL). Additionally, USDA257 likely produces a long-chain AHL with minimal mobility in TLC. Interestingly, the AHL profile obtained herein for SMH12 is perfectly consistent with existing knowledge regarding this strain of *S. fredii* [23]. The observation that no molecules of a different nature specifically vanish or change in the presence of c-di-GMP suggests that elevated concentrations of this metabolite induce a general impact on the quorum sensing (QS) state of these rhizobial populations, without specific interference with the various QS systems present within the cell.

### 3.2. High Levels of Cyclic di-GMP Diferentially Affect Autoagregation and Adhesion to Solid Surfaces in S. fredii

To elucidate additional roles of the second messenger c-di-GMP in *S. fredii*, we investigated its growth in diverse culture media commonly employed for assessing differential exopolysaccharide production in rhizobia. When cultured in rich TY medium, the USDA257-PleD and HWG35-PleD strains developed an autoaggregative phenotype characterized by floc formation, with this behavior being considerably more pronounced in HWG35-PleD (Figure 2A). As previously described for other *S. fredii* strains [39], the addition of genistein decreased the aggregative phenotype of these PleD strains grown in this rich medium. For the corresponding empty transposon versions (wild-type versions), USDA257 and HWG35 exhibited increased adhesion to the test tube walls at the air–liquid interface of the culture medium, as opposed to a purely autoaggregative phenotype. Conversely, SMH12 demonstrated reduced adhesion to the test tube walls in this area compared to SMH12-PleD, with no observation of floc formation or an autoaggregative phenotype in any tested conditions (Figure 2A).

In addition to the rich TY medium, bacterial growth was assessed in YM medium, where mannitol serves as the exclusive carbon source. Under these conditions, bacterial adhesion to the test tube walls at the air-liquid interface was not observed. Moreover, the visual inspection of the bacterial cultures indicated a qualitative increase in floc formation in the USDA257-PleD and HWG35-PleD than that seen in TY medium. The inclusion of genistein in the medium attenuated even more the formation of these flocs, which was inferred from the increased culture turbidity and reduced floc size (Figure 2A). However, the SMH12-PleD strain showed minimal growth in this medium, especially in the absence of inducing-flavonoids (Figure 2A).

A Congo red (CR) accumulation assay was also conducted on solid TY medium, as the results showed clearer differences compared to that obtained in YM cultures. CR is a dye known to bind to β-(1,4) linkages of D-glucopyranosyl units from surface polysaccharides and amyloid-type proteins. In this sense, this dyer has frequently been used when investigating polysaccharide mutant strains due to its capacity to bind many types of bacterial polysaccharides [40,41]. In all three strains, high c-di-GMP levels led to increased dye accumulation compared to physiological conditions, indicating altered exopolysaccharide production in the presence of this second messenger. Furthermore, the USDA257-PleD and HWG35-PleD strains developed a wrinkly phenotype, and some white colonies, likely suppressors of the phenotype, appeared without accumulating CR. Interestingly, the SMH12-PleD strain accumulated significantly more CR than the other strains without displaying a wrinkly phenotype. However, this accumulation was reduced to levels similar to the other strains when genistein was added to the medium, which simultaneously increased its roughness (Figure 2B). Collectively, these findings suggest that elevated c-di-GMP concentrations alter the composition of exopolysaccharides, thereby impacting autoaggregation and adhesion to solid surfaces, especially in the strains USDA257 and HWG35 of *S. fredii*.

### 3.3. Biofilm Formation Is Differentially Regulated by Elevated Cyclic di-GMP Levels Across S. fredii Strains

The production of extracellular polysaccharides is closely related to biofilm formation, due to their participation in the formation of the extracellular matrix [42]. Previous research demonstrated that the overexpression of diguanylate cyclases such as PleD significantly contributes to biofilm formation in bacteria of different genera [17,43]. Consistent with these observations, both the USDA257-PleD and HWG35-PleD strains of *S. fredii* exhibited a significant enhancement in their capacity for biofilm formation relative to their wild-type counterparts in TY medium. USDA257-PleD displayed values 12 to 18-fold higher than its wild-type strain. The HWG35-PleD strain showed the most pronounced effect, with biofilm formation often exceeding measurable limits due to staining-associated loss. Quantification from diluted samples indicated an increase in approximately 50-fold relative to the control (Figure 3). Conversely, the SMH12 strain showed the opposite trend. While some bacterial adhesion to the tube walls was observed for SMH12-PleD during self-aggregation assays, crystal violet quantification in microtiter plates revealed a drastic 10-fold decrease in biofilm formation in TY medium compared to the wild-type strain. Significantly, genistein diminished biofilm formation solely in the wild-type SMH12 strain, corroborating earlier studies on this specific strain [25] (Figure 3).

### 3.4. High Levels of AHLs and Cyclic di-GMP Activate the Expression of the S. fredii USDA257 T6SS Genes

The role of the T6SS in how bacteria interact with their environment is crucial, both for promoting interspecies competitiveness among bacteria and contributing to the pathogenesis of bacteria in their eukaryotic hosts [44,45]. Although information is limited, recent reports indicate that T6SS also plays a key role in symbiotic relationships between bacteria and their hosts, which has been especially evident in the specific symbiosis between rhizobia and legumes [11,46]. To determine whether the expression of this protein secretion system is regulated by the signaling molecules studied in this work, we selected the *S. fredii* USDA257 strain, which possesses a functional T6SS [11]. Although the SMH12 sequence has recently been deposited in GenBank (GCA_024400375.1) and this strain also possesses all genes required for a functional T6SS, we opted not to include it in this section due to insufficient background information.

To analyze T6SS expression in USDA257, we introduced the plasmid pMUS1483 (containing a transcriptional fusion of the *ppkA* gene promoter and the *lacZ* reporter gene) into both the wild-type and PleD strains of USDA257. The threonine kinase/phosphatase pair encoded by the genes *ppkA*-*pppA* is commonly used to infer T6SS expression, making its promoter region suitable for assessing the system’s functionality [11,47]. Additionally, to assess the influence of QS systems in the expression of the T6SS, the USDA257 strains harboring pMUS1483 were incubated for 24 or 72 h in the presence of supernatant from 6-day-old USDA257 cultures (10% *v*/*v*), which contain AHLs (Figure 1). In our previous report [11], we characterized the kinetics of *ppkA* expression by β-galactosidase assays, demonstrating phase-dependent activation with maximal expression in older cultures (54 h). Accordingly, we selected two representative time points: 24 h (late exponential phase, showing low *ppkA* activity) and 72 h (late stationary phase, beyond our previous results). After 24 h, the presence of supplementary AHLs did not significantly increase *ppkA* gene expression in both wild-type and PleD USDA257 strains. However, the high concentration of c-di-GMP in the USDA257-PleD strain led to double their β-galactosidase activity, indicating that the T6SS of USDA257 is positively controlled by high concentrations of this second messenger (Figure 4). Intriguingly, exogenous AHL supplementation of 72-h cultures (representing stationary growth phase) led to a notable increase in *ppkA* promoter activation. This enhancement was evident in both wild-type and PleD USDA257 strains, yielding significantly higher values than in their un-supplemented homologous cultures (Figure 4). This suggests a synergistic effect between high levels of AHLs and c-di-GMP for the activation of this protein secretion system in USDA257.

## 4. Discussion

In recent years, the crucial role of c-di-GMP as a master molecule orchestrating pivotal physiological processes that enable soil bacteria to adapt to complex and dynamic environments has gained significant attention. Elevated levels of c-di-GMP typically promote a transition from a motile, planktonic lifestyle to a sessile, surface-associated state, primarily by stimulating biofilm formation and repressing motility [48]. This molecular switch is essential for soil bacteria to establish persistent colonies, compete for resources, and engage in interactions with plant roots, directly influencing plant health and the broader soil ecosystem function [49]. The regulatory network of c-di-GMP is finely tuned by the balanced activities of diguanylate cyclases (DGCs) and phosphodiesterases (PDEs), allowing the integration of diverse environmental cues and enhancing bacterial adaptability [50]. Receptors for c-di-GMP, including various proteins and riboswitches, further transduce these signals to regulate downstream processes such as exopolysaccharide production, aggregation, virulence and assembly of protein secretion systems [12]. For example, in *Pseudomonas putida*, c-di-GMP can repress the T6SS via the FleQ-FleN complex, and in *Vibrio cholerae*, a riboswitch in the *tfoY* mRNA is controlled by c-di-GMP, preventing translation of T6SS mRNAs at high concentrations [51]. While extensive information exists regarding c-di-GMP influence on lifestyle changes in pathogenic bacteria and, more recently, in beneficial plant-interacting bacteria [52], its precise role in symbiotic bacteria like rhizobia and their lifestyle transitions remains less understood. Additionally, group behaviors like plant colonization must be coordinated by mechanisms such as QS, which regulates gene expression in response to population density via autoinducer molecules like AHLs. In this sense, *S. fredii* SMH12 forms symbiotic biofilms, which are crucial for colonizing soybean roots and establishing symbiosis, in a process controlled by QS and plant-secreted flavonoid inducers [25].

In the present study, we aimed to analyze the role of these key messengers in the initial stages of legume colonization by different *S. fredii* strains. The diversity of molecules and proteins involved in successful symbiotic interactions, such as extracellular polysaccharides, effector proteins, and lipo-chitooligosaccharides, necessitates their synthesis to be orchestrated by higher-level regulatory mechanisms. These mechanisms likely involve regulatory genes whose functions are controlled by key messengers interconnected within a complex network, closely tied to the bacteria metabolic fitness, such as c-di-GMP and AHLs.

To explore these intricate regulatory networks, we engineered various agriculturally relevant *S. fredii* strains (USDA257, HWG35, and SMH12) to constitutively express high levels of c-di-GMP using the mini-Tn7*pleD** transposon. A fundamental characteristic associated with artificially elevated c-di-GMP levels is the increased production of surface polysaccharides, which can be monitored by the accumulation of Congo red dye, known to bind to β-(1,4) linkages of D-glucopyranosyl units and amyloid-type proteins [40,41]. As expected, all three modified *S. fredii* strains showed strong accumulation of this dye with high levels of c-di-GMP, indicating altered exopolysaccharide production. This observation was further supported by the evident autoaggregation of the bacteria, forming large flocs, particularly pronounced in YM medium. The inclusion of genistein, partially repressed the excessive bacterial mucosity even at high c-di-GMP levels, which could otherwise hinder initial plant root colonization [39]. Notably, the SMH12-PleD strain exhibited residual growth in YM medium that was only partially restored with the addition of genistein as in other tested strains. On the other hand, we observed that PleD sinorhizobial strains adhered better to the tube walls when culturing in rich TY medium, a well-documented phenomenon for elevated c-di-GMP levels in many bacteria [53].

There is growing evidence of cross-talk between QS and c-di-GMP signaling pathways [54,55]. For instance, in the model bacterium *S. meliloti* 8530, artificially elevated c-di-GMP levels produced by synthetic expression of PleD led to the identification of a novel linear mixed-linkage (1→3)(1→4)-β-glucan. This process involved the coordinated regulation by both c-di-GMP and the ExpR/*SinI* QS system [18]. Although the existence of this regulatory intersection is supported by indirect evidence, the specific c-di-GMP-binding protein(s) that likely repress *sinI* transcription, thereby linking external signals to changes in intracellular c-di-GMP concentrations and ultimately modulating QS in *S. meliloti*, remains to be fully elucidated [15]. Our findings in *S. fredii* strains indicate that high c-di-GMP levels led to a significant decrease in AHL production, almost completely abolishing it in SMH12-PleD. The dramatic AHL reduction in SMH12-PleD, which prevented mature biofilm formation in microtiter plates despite stronger initial adhesion, might be due to its increased exopolysaccharide production. This supports earlier findings that SMH12 biofilm formation is coordinated by its QS systems [25]. In contrast, in USDA257-PleD and HWG35-PleD, the reduction in AHL concentrations, triggered by high c-di-GMP levels, leads to significantly higher biofilm formation compared to their wild-type counterparts, which is in agreement with the increased adhesion to the test tube walls at the air-liquid interface in these two strains. This suggests that c-di-GMP modulates QS-regulated behaviors like biofilm development and exopolysaccharide secretion in *S. fredii*. This intricate interaction might allow rhizobia to fine-tune their lifestyles in response to both external environmental signals and internal cellular status, thereby supporting robust communal adaptations.

The T6SS is widely recognized as a crucial mechanism for bacteria to interact with their environment, involved in both interspecies competition and pathogenesis towards eukaryotic hosts [5]. Recent reports highlight its increasing recognition in symbiotic relationships between bacteria and their hosts, particularly in the rhizobia-legume symbiosis [7,8,9,10,11]. Our study specifically investigated the T6SS of *S. fredii* USDA257, which possesses a single T6SS cluster that belongs to phylogenetic Group 3, one of the most prevalent groups within the Rhizobiales order. This strain induces its T6SS in nutrient-limited minimal media and during the stationary phase of growth, which points to a QS-mediated regulation [11]. The regulation of T6SS by QS has been observed in various eukaryotic host-interacting microorganisms like *Vibrio* spp., *Aeromonas hydrophila*, *Chromobacterium violaceum*, *Enterobacter cloacae*, and *P. aeruginosa* [51]. Our results demonstrate that high c-di-GMP levels are sufficient to activate T6SS expression in rich TY medium in *S. fredii*. Interestingly and in agreement with previous reports, the addition of AHLs shows a synergistic role with high c-di-GMP levels, which, although responsible for reducing endogenous AHL production, in combination with exogenous AHLs is able to reach maximum T6SS expression, possibly compensating for the lower AHL production of PleD strains. These data strongly suggest that gene processes regulated by QS and by c-di-GMP are interconnected.

Overall, the results obtained in this work indicate that, despite being phylogenetically closely related strains, they behave in a subtly different way in the modulation that c-di-GMP exerts on the synthesis of AHLs involved in biofilm formation. Furthermore, the regulation of T6SS appears to be mediated by c-di-GMP in USDA257, adding new pieces to the complex regulatory puzzle in which this ubiquitous molecule participates. The data presented herein establish the foundation for a novel research avenue, raising numerous questions for future investigation. Considering the potential link between c-di-GMP, AHLs, and rhizobial protein secretion systems, our aim is to continue characterizing the T6SS and its effectors in both USDA257 and SMH12. Furthermore, we plan to examine if alternative bacterial configurations, arising from varying concentrations of these key messengers (due to different surface polysaccharides or distinct T6SS effectors), could enhance symbiotic performance and modify the rhizobial host range.

## Figures and Tables

**Figure 1 microorganisms-13-02232-f001:**
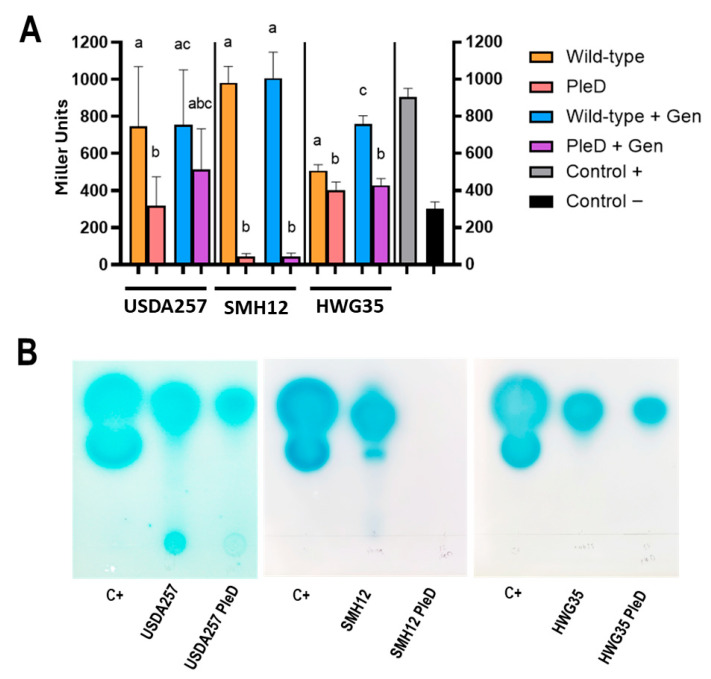
Influence of cyclic di-GMP on AHL production in *Sinorhizobium fredii*. (**A**) β-galactosidase activity measured using a modified assay with *A. tumefaciens* NT1 (pZLR4) as a biosensor, for AHL levels in the supernatants of Wild-type and PleD strains (mini-Tn7 and mini-Tn7-*pleD**, respectively) incubated or not with genistein (Gen). Data are presented as the average of three technical replicates per condition (±SD) in Miller units. Groups labeled with the same letter are not significantly different at α = 5% (One-Way ANOVA with multiple comparisons, *p* < 0.05). Commercial C6-HSL diluted in fresh culture was used as positive control, whereas uninoculated culture was used as negative control. (**B**) Thin layer chromatography (TLC) analysis of *N*-acyl-homoserine lactones (AHLs) produced by the wild-type and PleD version of the three *S. fredii* strains. Commercial C6-HSL and C8-HSL were used as controls. TLC were developed with the biosensor strain *A. tumefaciens* NT1 (pZRL4).

**Figure 2 microorganisms-13-02232-f002:**
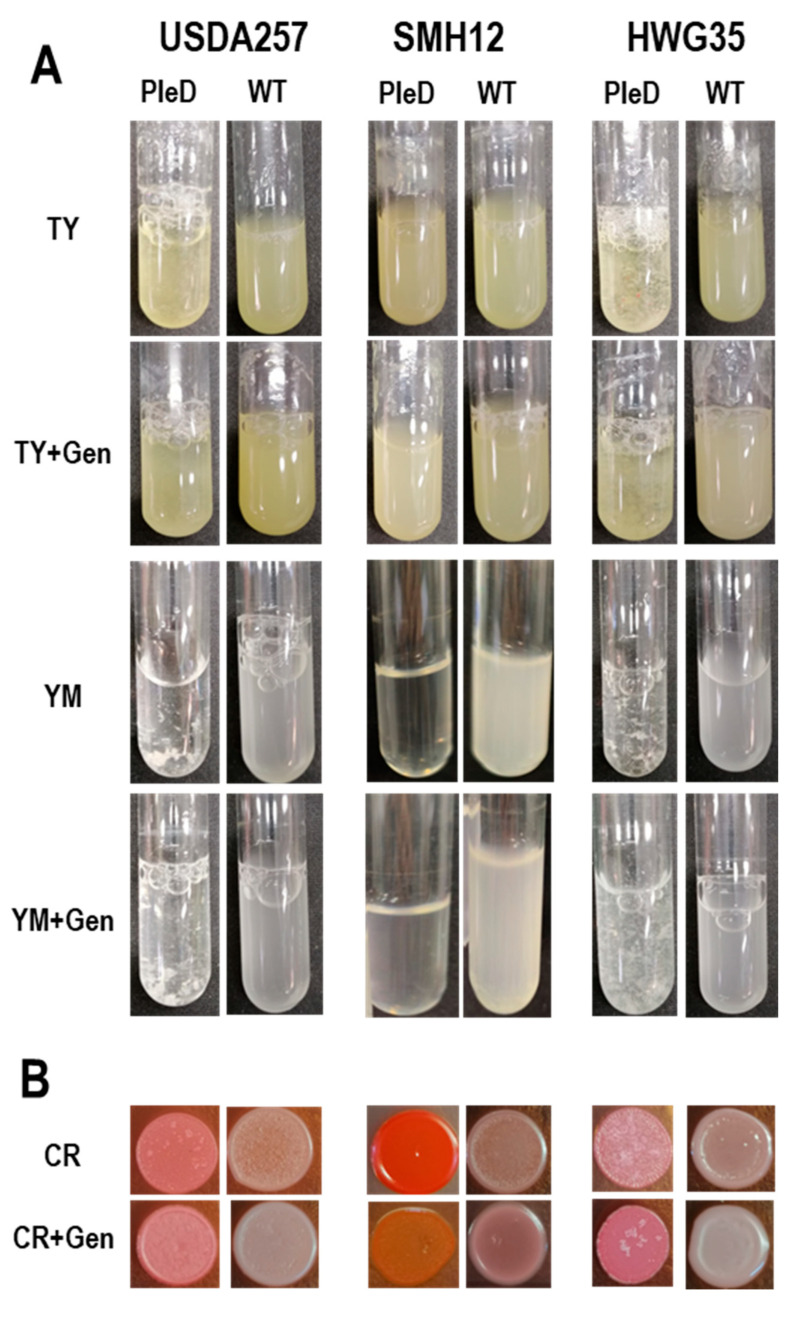
Bacterial autoaggregation and Congo red (CR) accumulation. (**A**) Differential growth of *S. fredii* strains in TY, YM in the presence or absence of genistein (Gen) for wild-type (WT) and PleD strains (mini-Tn7 and mini-Tn7-*pleD**, respectively). (**B**) Congo red accumulation of these *S. fredii* strains in TY medium containing or not the dye.

**Figure 3 microorganisms-13-02232-f003:**
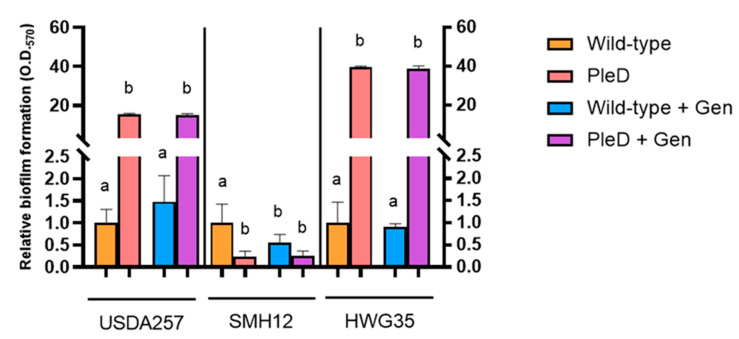
Biofilm formation under high levels of cyclic di-GMP in TY medium. Biofilm formation was quantified by staining with crystal violet and measuring the absorbance at 570 nm. Represented values are relative to the Wild-type and PleD strains (mini-Tn7 and mini-Tn7-*pleD**, respectively) grown in the absence or presence of genistein (Gen). Data are presented as the mean of three technical replicates per condition (±SD) in Miller units. Groups labeled with the same letter are not significantly different at α = 5% (One-Way ANOVA with multiple comparisons, *p* < 0.05).

**Figure 4 microorganisms-13-02232-f004:**
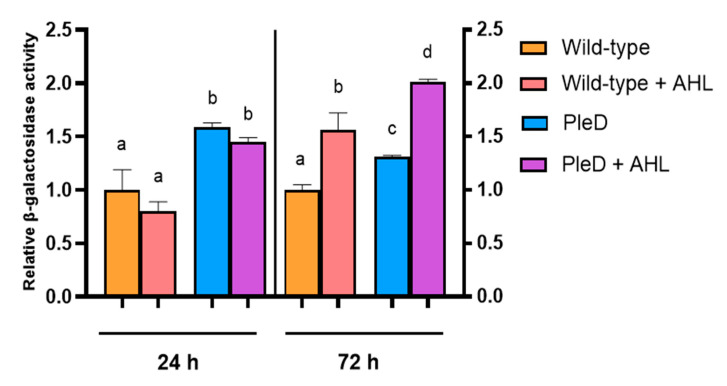
Activation pattern of the *S. fredii* USDA257 T6SS. Fold-change values of the β-galactosidase activity of USDA257 strains carrying a pMP220 plasmid containing the *ppkA* promoter region fused to the *lacZ* gene (pMUS1483) with respect to those values obtained by the wild-type strain. The tested strains included the Wild-type and PleD strains (mini-Tn7 and mini-Tn7-*pleD**, respectively) grown in the absence or presence of additional AHLs, all of them harboring the plasmid pMUS1483. Expression data are presented as the mean (±standard deviation). Groups labeled with the same letter are not significantly different at α = 5% (One-Way ANOVA with multiple comparisons, *p* < 0.05).

**Table 1 microorganisms-13-02232-t001:** Bacterial strains and plasmids used in this study.

Strain or Plasmid	Relevant Characteristics	Reference
*S. fredii*		
USDA257	Wild-type strain, Rif^R^	[32]
USDA257-miniTn7Km	USDA257-miniTn7, Km^R^	This work
USDA257-PleD	USDA257-miniTn7*pleD**, Km^R^	This work
USDA257-miniTn7Km pMUS1483	USDA257-miniTn7, Km^R^, containing a plasmid with a P*ppkA*::*lac*Z fusion. See below	This work
USDA257-PleD pMUS1483	USDA257-miniTn7-*pleD**, Km^R^ containing a plasmid with a P*ppkA*::*lac*Z fusion. See below	This work
SMH12	Wild-type strain, Rif^R^	[33]
SMH12-miniTn7Tc	SMH12-miniTn7Tc^R^	This work
SMH12-PleD	SMH12-miniTn7*pleD**, Tc^R^	This work
HWG35	Wild-type strain, Rif^R^	[26]
HWG35-miniTn7-Km	HWG35-miniTn7-Km^R^	This work
HWG35-PleD	HWG35-miniTn7-*pleD**, Km^R^	This work
*A. tumefaciens*		
NT1 (pZRL4)	*A. tumefaciens* devoid of pTiC58 and harboring pZRL4, which carries the fusion *traG*::*lacZ* and the *traR* gene	[34]
*E. coli*		
β2163	MG1655::Δ*dapA*::(erm-pir)RP4-2, Tc::Mu, Km^R^, Em^R^	[35]
β2155	RP4-2-Tc::Mu Δ*dapA*::(erm-pir) *thrB1004*, *pro*, *thi*, *strA*, *hsdS*, *lacZ* ΔM15, (F′ l*acZ* ΔM15, *lacIq*, *traD36*, *proA+*, *proB+*) Km^R^, Sm^R^, Em^R^	[35]
Plasmids		
mini-Tn7*pleD**-Km	mini-Tn7*pleD** containing Km marker Ap^R^, Km^R^	[17]
mini-Tn7Km	mini-Tn7*pleD**Km with an internal deletion of *pleD** Ap^R^, Km^R^	[17]
mini-Tn7*pleD**Tc	mini-Tn7*pleD** containing Tc marker Ap^R^, Tc^R^	[17]
mini-Tn7Tc	mini-Tn7*pleD**Tc with an internal deletion, Ap^R^, Tc^R^	[17]
pUX-BF13	Helper plasmid providing the Tn7 transposition functions in trans, Ap^R^, *mob*+, ori-R6K	[36]
pMUS1483	*S. fredii* USDA257 *ppkA* promoter region cloned into pMP220 as a transcriptional fusion, P*ppkA*::*lac*Z, RK2 ori, Tc^R^. [pMP220::P*ppkA*]	[11]

## Data Availability

The raw data supporting the conclusions of this article will be made available by the authors on request.

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
