# Peer review of "Cyclic di-GMP Modulation of Quorum Sensing and Its Impact on Type VI Secretion System Function in Sinorhizobium fredii"

_microorganisms, 2025, doi:10.3390/microorganisms13102232_

Round 1

Reviewer 1 Report

Comments and Suggestions for Authors

The manuscript under review presents interesting research results on the symbiosis between rhizobia and leguminous plants, aiming to provide insights into the mechanisms regulating the relationship between plants and bacteria. What are the results of this research? I want to ask the Authors to specify how they see the practical application of the obtained research results. In plant breeding? In agricultural production? In other areas?

Comments

Introduction

It provides a good introduction to the research problem.

Materials and Methods

In which year was the research conducted? Where?

Results

The figures and tables are clear and necessary.

Discussion

It is quite short.

Conclusions?

References

Quite a large number of publications (56). Please limit yourself to the most recent publications. Please remove older publications, especially those from the 20th century.

Author Response

The manuscript under review presents interesting research results on the symbiosis between rhizobia and leguminous plants, aiming to provide insights into the mechanisms regulating the relationship between plants and bacteria. What are the results of this research? I want to ask the Authors to specify how they see the practical application of the obtained research results. In plant breeding? In agricultural production? In other areas?

Thanks for this observation. In our opinion, this work is not focused on a short-term practical application since all the agronomical interesting rhizobial strains genetically engineered, are not allowed to be released into field in order to check its symbiotic performance. However, all the information collected on the complex genetic regulation of the rhizobial symbiotic interaction, may help in the finding of natural isolated strains that hypothetically could be used as natural inoculants. For instance, HWG35 is a highly competitive strain that was isolated from Chinese soils by our group more than 20 years ago. Why is so competitive in nature? Maybe for its natural capacity to form biofilms as we report? At this moment we do not have its DNA sequence, but maybe it has a T6SS that can help to overcompete other rhizobia in soils. Therefore, even though that genetic modifications will not be used in soil plant tests, they are mandatory to explain some unknown concerns as strain competitiveness.

Comments

Introduction

It provides a good introduction to the research problem.

Thanks!

Materials and Methods

In which year was the research conducted? Where?

This research was done during 2025 in the department of Microbiology of Biology Faculty in the University of Seville (Spain)

Results

The figures and tables are clear and necessary.

Thanks!

Discussion

It is quite short.

We have now included more information in the discussion and since we opted for do not include a conclusion section as stated below; we believe that now this section has been improved.

Conclusions?

Thank you for your suggestion. As stated in the instructions for authors the Conclusion section is not mandatory but can be added to the manuscript if the discussion is unusually long or complex”), we consider that our discussion does not fall into this category. Therefore, we have opted to present the conclusions within the discussion itself. However, we have now highlighted some points more explicitly to address the reviewers’ concerns.

References

Quite a large number of publications (56). Please limit yourself to the most recent publications. Please remove older publications, especially those from the 20th century.

We appreciate the reviewer's comment; we only could eliminate one publication following the comment of other reviewer. However, we believe that we have referenced adequately throughout the text. Furthermore, while it is true that some of our references are a bit old, the majority of them describe media and/or protocols that have been used in rhizobiology for decades, so we cannot remove them.

Reviewer 2 Report

Comments and Suggestions for Authors

   - 2.1 In "after several attemps", the word "attemps" is misspelled and should be corrected to "attempts".  
   - 2.2 In "A 4 μL (2+2) mixture... weas used", the word "weas" is misspelled and should be corrected to "was".  
      - 3.1 In "the addition of genistein also shows a different behavior", there is a tense error. Consistent with other experimental result descriptions, the present tense "shows" should be changed to the past tense "showed".  

- 3.4 In "leaded to double their β galactosidase activity", the verb "leaded" is grammatically incorrect and should be replaced with "led".  
   - The legend of Figure 2A, "in presence or absence of genistein (Gen) of wild-type", contains grammatical confusion and should be revised to "in the presence or absence of genistein (Gen) for wild-type".  

The title of Figure 4, "Acivation pattern of the s.freif ADA27 76S5", is incorrect and should be corrected to "Activation pattern of the S. fredii USDA257 T6SS".  

For Reference 16 (Escobar, M.R. et al.), the publication year is missing. Regarding Reference 26, its research object is Sinorhizobium fredii HH103, but it is cited in the section discussing S. fredii SMH12 in the text. Is there a risk of incorrect citation here?  

In the TLC analysis (Section 2.2):  
   - No literature support is provided for selecting "C6-HSL and C8-HSL as controls".  
   - For the elution system "methanol:water (60:40 v/v)", it is necessary to clarify whether this ratio is the result of optimization and explain the reasons for choosing this ratio.  

In Section 2.4:  
   - In "using Synergy HT microplate reader (BioTec, USA)", it is necessary to confirm whether "BioTec" is a misspelling of the brand name "BioTek". Additionally, further details on the detection parameters (e.g., temperature, oscillation conditions) should be provided.  
   - For "washing three times" after crystal violet staining, more detailed conditions (e.g., volume of washing solution, soaking or centrifugation time) should be specified.  

In Section 3.1, the statement "Most remarkably, SMH12-PleD showed virtually no production" is not scientifically precise. It is necessary to clarify whether "virtually no production" means the target substance (AHLs) was undetectable by the used detection method or if its concentration was below the detection limit.  

In Section 3.2, the description "Moreover, floc formation in the USDA257-PleD and HWG35-PleD strains was considerably greater than that seen in TY medium" is only qualitative. A scientific quantitative approach (e.g., measuring floc average diameter, floc count per milliliter of culture, or the proportion of flocs relative to total bacterial biomass) should be adopted, and statistical significance of differences between groups should be analyzed and reported.  

In Section 3.3, the statement mentions "The HWG35-PleD strain showed the most pronounced effect, with biofilm formation often exceeding measurable limits due to loss during the staining process (about 50-fold higher)". If biofilm formation exceeded the measurable range, it is necessary to explain how the "50-fold higher" result was derived (e.g., whether it was calculated based on diluted samples or extrapolated from standard curves).  

In Section 3.4, the result states "Intriguingly, exogenous AHL supplementation of 72-hour cultures (representing stationary growth phase) led to a notable increase in ppkA promoter activation". It is necessary to clarify:  
    - The reason for using cultures of different time points (24 h and 72 h) in the experiment.  
    - Whether both 24-hour and 72-hour cultures were in the stationary growth phase (supporting data such as bacterial growth curves should be provided if possible).  
    - Whether differences in bacterial growth phases could affect the detection results of ppkA promoter activation.  

The Discussion section requires further in-depth analysis:  
    - Although the text repeatedly mentions "the addition of AHLs shows a synergistic role with high c-di-GMP levels", the underlying molecular mechanisms are not discussed in depth. For example: Does c-di-GMP affect T6SS activity by binding to T6SS regulatory proteins? Do AHLs indirectly regulate c-di-GMP metabolism through quorum sensing genes (e.g., *sinI*/*sinR*)? Currently, the discussion only describes phenomena without exploring the crosstalk nodes between signaling pathways.  
    - When discussing "Notably, the SMH12-PleD strain exhibited residual growth in YM medium" and speculating that "the strain may synthesize additional extracellular polysaccharides", both experimental evidence (e.g., Congo red staining results for SMH12-PleD) and relevant literature support should be provided to validate this hypothesis.

Author Response

   - 2.1 In "after several attemps", the word "attemps" is misspelled and should be corrected to "attempts".

Done

     - 2.2 In "A 4 μL (2+2) mixture... weas used", the word "weas" is misspelled and should be corrected to "was".

Done  
      - 3.1 In "the addition of genistein also shows a different behavior", there is a tense error. Consistent with other experimental result descriptions, the present tense "shows" should be changed to the past tense "showed".  

Done  

- 3.4 In "leaded to double their β galactosidase activity", the verb "leaded" is grammatically incorrect and should be replaced with "led".

Done  

   - The legend of Figure 2A, "in presence or absence of genistein (Gen) of wild-type", contains grammatical confusion and should be revised to "in the presence or absence of genistein (Gen) for wild-type".  

Done  

The title of Figure 4, "Acivation pattern of the s.freif ADA27 76S5", is incorrect and should be corrected to "Activation pattern of the S. fredii USDA257 T6SS".  

We have revised it, but in the last version that we downloaded it is correct. We do not know why the version that was reviewed appears like this. But could you please double-check it just in case? Thanks.

For Reference 16 (Escobar, M.R. et al.), the publication year is missing.

Included (2023), sorry for that.

Regarding Reference 26, its research object is Sinorhizobium fredii HH103, but it is cited in the section discussing S. fredii SMH12 in the text. Is there a risk of incorrect citation here?  

You are right, thank you for your observation. The mistake has an explanation, since this reference is cited in the introduction as our last paper in which we worked with QS signaling, but with other bacteria that we use in our lab as you point out (HH103). We have removed this paper from the reference list.

In the TLC analysis (Section 2.2):

  - No literature support is provided for selecting "C6-HSL and C8-HSL as controls".

Thanks for your appreciation. These are the controls that we routinely use in our lab with our strains. We have previously published TLCs pics containing these controls (in reference 24 figure 3). Therefore, we have included this reference as you proposed.

  - For the elution system "methanol:water (60:40 v/v)", it is necessary to clarify whether this ratio is the result of optimization and explain the reasons for choosing this ratio.  

The TLC analysis for N-Acyl homoserine lactones is well established from its first publication (Shaw et al., 1997 PNAS doi: 10.1073/pnas.94.12.6036), where the authors described the method with this ratio. We included reference 23 from 2013 that is the first one of our lab in which we carried out this technic. According to other reviewers’ suggestions, we decided to include recent publications instead of the classic one of Shaw 1997.

In Section 2.4:

     - In "using Synergy HT microplate reader (BioTec, USA)", it is necessary to confirm whether "BioTec" is a misspelling of the brand name "BioTek". Additionally, further details on the detection parameters (e.g., temperature, oscillation conditions) should be provided.  

We have changed to BioTek, thanks. Regarding the detection parameters, we do not include anyone since no oscillation was used, and the temperature was not modified from the setting. We have included the sentence “using standard parameters”.

   - For "washing three times" after crystal violet staining, more detailed conditions (e.g., volume of washing solution, soaking or centrifugation time) should be specified.

It has been included now.

In Section 3.1, the statement "Most remarkably, SMH12-PleD showed virtually no production" is not scientifically precise. It is necessary to clarify whether "virtually no production" means the target substance (AHLs) was undetectable by the used detection method or if its concentration was below the detection limit.  

In these experiments, we could detect around 50-90 Miller units in SMH12-PleD, below the negative control. For that reason, we included that sentence but now we have changed to percentage reduction as in other strains. “SMH12-PleD showed a reduction around 95% in AHL production”

In Section 3.2, the description "Moreover, floc formation in the USDA257-PleD and HWG35-PleD strains was considerably greater than that seen in TY medium" is only qualitative. A scientific quantitative approach (e.g., measuring floc average diameter, floc count per milliliter of culture, or the proportion of flocs relative to total bacterial biomass) should be adopted, and statistical significance of differences between groups should be analyzed and reported.  

We agree that the observation of increased flocculation is exclusively qualitative and not quantitative. A quantitative approach would require additional experiments. However, this would take more time than the 10 days the journal has given us for the article's revision. Furthermore, we believe that quantitative microtiter plate adhesion experiments better reflect how the entire biofilm formation process is affected in each strain. In any case, we have rephrased the sentence to clarify that it is merely a visual observation.

In Section 3.3, the statement mentions "The HWG35-PleD strain showed the most pronounced effect, with biofilm formation often exceeding measurable limits due to loss during the staining process (about 50-fold higher)". If biofilm formation exceeded the measurable range, it is necessary to explain how the "50-fold higher" result was derived (e.g., whether it was calculated based on diluted samples or extrapolated from standard curves).  

Thanks for your observation. This observation was done by diluting the overstained biofilm and performing the calculation. We have included it in the text.

 In Section 3.4, the result states "Intriguingly, exogenous AHL supplementation of 72-hour cultures (representing stationary growth phase) led to a notable increase in ppkA promoter activation". It is necessary to clarify:

- The reason for using cultures of different time points (24 h and 72 h) in the experiment.

- Whether both 24-hour and 72-hour cultures were in the stationary growth phase (supporting data such as bacterial growth curves should be provided if possible).  
 - Whether differences in bacterial growth phases could affect the detection results of ppkA promoter activation.  

We have included in the text some of following information that we consider more important. In our previous work (Reyes-Perez et al., 2025; Ref. 11), we characterized the kinetics of ppkA expression by β-galactosidase assays, demonstrating phase-dependent activation with maximal expression in older cultures (54 h). Accordingly, we selected two representative time points: 24 h (late exponential phase, showing low ppkA promoter activity) and 72 h (late stationary phase, beyond our previous results), in line with standard growth curves for Sinorhizobium fredii. These growth phases are well established for fast-growing rhizobia for more than 20 years. Thus, the rationale for using different culture times is to evaluate AHL effects under distinct physiological states. As reported previously (ref 11), ppkA expression varies with bacterial growth phase, confirming that detection differences mirror regulatory biology, not technical artifacts.

The Discussion section requires further in-depth analysis:

     - Although the text repeatedly mentions "the addition of AHLs shows a synergistic role with high c-di-GMP levels", the underlying molecular mechanisms are not discussed in depth. For example: Does c-di-GMP affect T6SS activity by binding to T6SS regulatory proteins? Do AHLs indirectly regulate c-di-GMP metabolism through quorum sensing genes (e.g., *sinI*/*sinR*)? Currently, the discussion only describes phenomena without exploring the crosstalk nodes between signaling pathways.

All these insightful observations shall be the subject of future specific studies. At present, however, we have not identified a specific regulatory protein in S. fredii that binds c-di-GMP and modulates T6SS expression. This question is being addressed in our ongoing RNA-seq projects, but currently, we lack definitive data to provide a conclusive answer. Regarding the crosstalk between quorum sensing (QS) and c-di-GMP signaling pathways, we have revised the discussion accordingly (lines 445-452). We hope that this updated version satisfactorily addresses the reviewer’s concerns.

    - When discussing "Notably, the SMH12-PleD strain exhibited residual growth in YM medium" and speculating that "the strain may synthesize additional extracellular polysaccharides", both experimental evidence (e.g., Congo red staining results for SMH12-PleD) and relevant literature support should be provided to validate this hypothesis.

Thank you for your comment. At this stage of our research, we do not have sufficient experimental evidence to confirm that the SMH12-PleD strain synthesizes additional extracellular polysaccharides compared to the wild type when grown in YM medium, responsible for limiting its growth. What we observe is that the SMH12-PleD strain exhibits slower growth in YM medium, and this impaired growth is partially alleviated by the addition of genistein, which is known to repress extracellular polysaccharide production (as shown in Figure 2). Accordingly, to avoid any misinterpretation, we have rephrased this section in the manuscript for greater clarity.

Reviewer 3 Report

Comments and Suggestions for Authors

The work investigated the regulatory interplay between AHLs, c-di-GMP, and their influence in the T6SS activity in three Sinorhizobium fredii strains of agronomic relevance. The work in general sounds good, but I am not very familiar with this area of research. There are some suggestions that can improve the work and increase the quality of the manuscript.

  1. Language and grammar of the manuscript should be improved throughout the text.
  2. The section of 'Abstract'should be rewritten, and described the main methods or treatments applied, and summarized the article's main findings.
  3. The last paragraph of the'Introduction' was suggested to deleted or put in the conclusion section.
  4. Add a conclusion section where to point each of your main findings, preferably with some values.
  5. Line 260, ‘in the absence of presence of genistein’, please check this sentence.
  6. In Figure 2B, what is RC? It was not introduced throughout the text.
  7. In Figure 3, it was stated ‘Data are presented as the mean of three technical replicates per condition (± SD) in Miller units.’However, in lines 197-198 of the section of ‘Materials and Methods’, it was stated that ‘Experiments were performed a minimum of three times, with each run including three biological replicates and two technical replicates for every condition. Why?
  8. Lines 340-350 of the section of ‘Results’, these contents can be put in the section of ‘Materials and Methods’.
  9. Reference style is inadequate, need to prepare in an unified form, please prepare according to the “Instructions for Authors”.

Author Response

The work investigated the regulatory interplay between AHLs, c-di-GMP, and their influence in the T6SS activity in three Sinorhizobium fredii strains of agronomic relevance. The work in general sounds good, but I am not very familiar with this area of research. There are some suggestions that can improve the work and increase the quality of the manuscript.

  1. Language and grammar of the manuscript should be improved throughout the text.

The text has been fully revised to correct all the detected mistakes.

2. The section of 'Abstract'should be rewritten, and described the main methods or treatments applied, and summarized the article's main findings.

Thanks for this point, the abstract has been restructured according to the reviewers concerns and we think that this new version will help readers to appreciate the relevance of our findings.

3. The last paragraph of the 'Introduction' was suggested to deleted or put in the conclusion section.

We have included it at the end of the discussion section as suggested to address points 3 and 4 of this reviewer, and similar concerns of other reviewers. A more general sentence serving as final remark of the introduction has been included.

4. Add a conclusion section where to point each of your main findings, preferably with some values.

Thank you for your suggestion. As stated in the instructions for authors the Conclusions section is not mandatory but can be added to the manuscript if the discussion is unusually long or complex, we consider that our discussion does not fall into this category. Therefore, we have opted to present the conclusions within the discussion itself. However, we have now highlighted some points more explicitly to address the reviewers’ concerns.

5. Line 260, ‘in the absence of presence of genistein’, please check this sentence.

The sentence is correct. The figure caption relates to the treatments including presence of genistein (+ gen) blue and purple colors, or physiological conditions (absence of genistein) orange or pink colors. For a better understanding we have change it to “incubated or not with genistein (Gen)”

6. In Figure 2B, what is RC? It was not introduced throughout the text.

Thanks for your appreciation. RC is actually CR, Congo Red what is stated in the figure caption (line 295) and just below the figure (line 299). It is now corrected.

7. In Figure 3, it was stated ‘Data are presented as the mean of three technical replicates per condition (± SD) in Miller units.’ However, in lines 197-198 of the section of ‘Materials and Methods’, it was stated that ‘Experiments were performed a minimum of three times, with each run including three biological replicates and two technical replicates for every condition. Why?

Thanks for this observation. The biofilm quantitation by crystal violet staining is a technic with some inherent variability, and therefore several replicates are mandatory. We use three biological replicates per experiment (that means independent clones) and two technical replicates (two measures in two independent wells of the same clone). This way we usually get reliable results to confirm the results as we usually report in our previous papers.

8. Lines 340-350 of the section of ‘Results’, these contents can be put in the section of ‘Materials and Methods’.

We are aware that this paragraph does not specifically present a result. However, given the limited information available regarding the role of T6SS in the rhizobium-legume relationship, we believe its inclusion here as a justification and introductory remark for section 3.4 is important. We consider that it is not necessary to include this paragraph in the materials and methods section, as it is only a consideration already described in the introduction (lines 105–121). We hope this explanation clarifies the importance of this section.

9. Reference style is inadequate, need to prepare in an unified form, please prepare according to the “Instructions for Authors”.

Reference style has been revised and unified.

Reviewer 4 Report

Comments and Suggestions for Authors

This study investigated the relationship between cyclic di-GMP and N-acyl homoserine lactones (AHLs) using different strains of Sinorhizobium fredii. It was clearly observed that an increase in cyclic di-GMP led to a reduction in AHL production, while also altering autoaggregation, adhesion, and biofilm formation. Additionally, high concentrations of c-di-GMP activated the expression of T6SS in USDA257, and exogenous AHL supplementation significantly increased the activation of the ppkA promoter, showing a synergistic effect with c-di-GMP. Although the results of this study are clear, some issues still need clarification:

  1. The abstract lacks a description of the key findings of this study. It should be rewritten to include the results, allowing readers to understand the significance of this work from the abstract.
  2. Line 176: A brief description of the β-galactosidase assay method should be provided.
  3. Line 226: The authors mention that "AHL production was reduced by 20% in HWG35-PleD and by 50% in USDA257-PleD." Is this based on the TLC analysis in Figure 1? How was the quantification performed?
  4. Figure 1: The significance markers in the figure appear incorrect, such as those for "USDA257 Wild-type + Gen" and "HWG35 Wild-type + Gen."
  5. Lines 229–232: The effect of genistein addition on AHLs is mentioned, but no corresponding data is shown.
  6. Figure 2B: The Congo red accumulation results are presented in an image that is too small, making it difficult to interpret the findings. It is recommended to present this as a larger, standalone figure for better clarity.
  7. Figure 4: The significance markers should be verified for accuracy. Typically, they should follow alphabetical order.

Author Response

This study investigated the relationship between cyclic di-GMP and N-acyl homoserine lactones (AHLs) using different strains of Sinorhizobium fredii. It was clearly observed that an increase in cyclic di-GMP led to a reduction in AHL production, while also altering autoaggregation, adhesion, and biofilm formation. Additionally, high concentrations of c-di-GMP activated the expression of T6SS in USDA257, and exogenous AHL supplementation significantly increased the activation of the ppkA promoter, showing a synergistic effect with c-di-GMP. Although the results of this study are clear, some issues still need clarification:

  1. The abstract lacks a description of the key findings of this study. It should be rewritten to include the results, allowing readers to understand the significance of this work from the abstract.

Thanks for this point, the abstract has been restructured according to the reviewers concerns and we think that this new version will help readers to appreciate the relevance of our findings.

2. Line 176: A brief description of the β-galactosidase assay method should be provided.

It is now included from lines 179 to 183.

3. Line 226: The authors mention that "AHL production was reduced by 20% in HWG35-PleD and by 50% in USDA257-PleD." Is this based on the TLC analysis in Figure 1? How was the quantification performed?

Our TLC analysis is intended to be qualitative only. The quantification refers to the reduction in average AHL production, which is estimated from the β-galactosidase activity data presented in Figure 1A. For instance, in USDA257 wt versus USDA257 PleD, the 50% decrease corresponds to a reduction from approximately 700 to 350 Miller Units. To improve clarity, we have included a reference to Figure 1A in line 236.

4. Figure 1: The significance markers in the figure appear incorrect, such as those for "USDA257 Wild-type + Gen" and "HWG35 Wild-type + Gen."

Thanks, the statistical test has been double checked and the markers are right, probably due to the experimental variability. As the test is distinct for each strain, the alphabetical order is independent on each strain.

5. Lines 229–232: The effect of genistein addition on AHLs is mentioned, but no corresponding data is shown.

The effect of genistein addition is included in the figure 1A, the blue bars for the wild type strains and the purple bars for the PleD strains. To upgrade the intelligibility, we also have included a reference to Figure 1A in the text (now line 240).

6. Figure 2B: The Congo red accumulation results are presented in an image that is too small, making it difficult to interpret the findings. It is recommended to present this as a larger, standalone figure for better clarity.

In order to maintain the proportions with Figure 2A and following the reviewers’ suggestions, we enlarged the entire Figure 2, which now fits the whole page including the figure legend.

7. Figure 4: The significance markers should be verified for accuracy. Typically, they should follow alphabetical order.

The markers have been verified, and they follow alphabetical order in each separate strain since they are independent statistical analysis, a and b in each strain.

Round 2

Reviewer 2 Report

Comments and Suggestions for Authors

This article can be published in its current form.

Author Response

Thank you

Reviewer 3 Report

Comments and Suggestions for Authors

The authors have made some detailed responses or modifications, however, I would like to ask the authors to supplement the methodological part with the following information:

1) provide the composition of the substrate or manufacturer

2) provide the accuracy of the devices 

Author Response

The authors have made some detailed responses or modifications, however, I would like to ask the authors to supplement the methodological part with the following information:

1) provide the composition of the substrate or manufacturer

We are not entirely certain what the reviewer intended by "substrate" and "manufacturer." Our interpretation is that this refers to the culture media, is that correct? If so, the culture media are appropriately referenced in the text to avoid unnecessary repetition. Moreover, the suppliers of the compounds required for media preparation differ across countries, and therefore are intrinsically dependent on the local context.

2) provide the accuracy of the devices 

Regarding the accuracy of the devices, we assume that the reviewers refers to Synergy HT microplate reader. Thus, for OD range 0.000 to 2.000 (96-well plates) the accuracy is ± 1.0% ± 0.010 OD.

Reviewer 4 Report

Comments and Suggestions for Authors

Thank you for your detailed response. Authors have addressed the questions I raised. However, the resolution of Figure 2B, the Congo red experiment, still needs improvement. The current image quality does not allow for a clear comparison with the results described in the text.

Author Response

Thank you for your detailed response. Authors have addressed the questions I raised. However, the resolution of Figure 2B, the Congo red experiment, still needs improvement. The current image quality does not allow for a clear comparison with the results described in the text.

We appreciate the reviewer’s comment. All figures have been replaced with high‑resolution versions, which we trust will improve the quality and clarity of the manuscript.